# A Saprophytic Fungus *Tubeufia rubra* Produces Novel Rubracin D and E Reversing Multidrug Resistance in Cancer Cells

**DOI:** 10.3390/jof9030309

**Published:** 2023-02-28

**Authors:** Shengyan Qian, Xuebo Zeng, Yixin Qian, Yongzhong Lu, Zhangjiang He, Jichuan Kang

**Affiliations:** 1Engineering and Research Center for Southwest Bio-Pharmaceutical Resources of National Education Ministry of China, Guizhou University, Guiyang 550025, China; 2School of Life Science, Guizhou University, Guiyang 550025, China; 3Guizhou Key Laboratory of Characteristic Microbial Research & Drug Development, Zunyi Medical University, Zunyi 563000, China; 4School of Food and Pharmaceutical Engineering, Guizhou Institute of Technology, Guiyang 550003, China

**Keywords:** *Tubeufia rubra*, secondary metabolites, Glyceroglycolipids, cancer multidrug resistance, reverse, P-glycoprotein

## Abstract

Two new (**1**, **2** viz Rubracin D and E) and sixteen known Glyceroglycolipids (**3**–**18**) in the saprophytic fungus *Tubeufia rubra* (PF02-2) from decaying wood in freshwater habitat were isolated and identified. Their chemical structures were elucidated via means of the extensive spectroscopic analyses of NMR, HR-ESI-MS and UV spectra, as well as comparison with literature data. The new compounds were assayed for the reversal activity of multidrug resistance (MDR) on MCF-7/ADM, K562/ADM and A549/ADM cell lines, and both compounds **1** and **2** reversed MDR in the three resistant cancer cell lines with concentration dependence. In the assay on K562/ADM, both new compounds had been proved to have remarkable MDR reversal effects, which were higher than those of the positive control viz Verapamil (Vrp). Meanwhile, in the assay on A549/ADM, compound **1** displayed significant MDR reversal effects, which were also higher than those of Vrp at certain concentrations. Furthermore, the Western blot assay proved that both new compounds reversed the MDR in the resistant cancer cell line viz MCF-7/ADM by inhibiting the overexpression of P-glycoprotein. This is the first report that the Glyceroglycolipids isolated firstly from the fungal genus *Tubeufia* reversed MDR in resistant cancer cells.

## 1. Introduction

Cancer is still a leading cause of death worldwide [1,2]. Multidrug resistance (MDR) in cancer cells is a key factor in malignancies treatment with chemotherapeutic drugs [3]. P-glycoprotein (P-gp), a 170-kDa transmembrane protein, plays an important role in the process of MDR with its overexpression. Searching for P-gp inhibitors that can be combined with anticancer drug therapies is one of the most efficient strategies to reverse cancer MDR [4]. Natural products are important bioactive agents for developing a new clinic drug because of their diverse structures and potential biological activities [5,6,7,8,9]. A large number of natural products have reported the potential to inhibit the overexpression of P-gp resulting in reversing MDR activity in tumors [10,11,12] with structures involved in alkaloids [13], flavonoids [14], lignans [15,16] and terpenoids [17,18].

To date, many bioactive secondary metabolites were reported from the microorganism, in 1993, Stierle et al. [19] found that the strain *Taxomyces andreanae* isolated from the phloem (inner bark) of *Taxus brevifolia* had the ability to produce taxol, and numerous natural products with complicated structure and novel activity [20,21,22]. For example, the esters of orsellinic acid derivatives globosumones A-C isolated from *Chaetomium globosum* exhibited significant cytotoxic activity against four cancer cell lines NCI-H460, MCF-7, SF-268, and WI-38 [23]. Two novel xanthone dimers phomoxanthones A and B obtained the endophytic fungus *Phomopsis* sp. BCC 1323 showed antimalarial and antitubercular activities [24]. A new chromone aspergilluone A produced by a marine sponge-associated fungus *Aspergillus* sp. LS57 displayed antimicrobial activity against *Staphylococcus aureus* and *Bacillus subtilis* [25]. The secondary metabolites from *Helicomyces* sp. No.19353 demonstrated significant antidiabetic activity [26]. Therefore, it is important to find active substances in microbial fermentation broth.

Our previous studies indicated that the crude ethyl acetate extract of the fermentation broth from the strain *Tubeufia rubra* (PF02-2) showed significant reversal activity of MDR in MCF-7/ADM cell line via inhibiting the overexpression of P-gp [27]. A novel nitrogen-containing compound Rubracin A with reversed MDR activity was isolated from the strain *Tubeufia rubra* (PF02-2) [28], while the other active compounds awaited clarification. In this study, the saprophytic fungus PF02-2 of the decaying wood in a freshwater habitat and the secondary metabolites from its fermented culture were further investigated. Herein, the isolation, structure elucidation and bio-activity assay of another new class of compounds from the crude ethyl acetate extract of PF02-2 were described in detail.

## 2. Materials and Methods

### 2.1. General Experimental Procedures

HR-ESI-MS was performed on a Waters Xevo G2 QTOF mass spectrometer. IR spectra were recorded with NICOLET iS10. UV spectra were obtained using a Shimadzu UV-2401A spectrophotometer. Optical rotations were measured on a JASCO-20C digital polarimeter. Then, 1D NMR and 2D NMR were performed on Bruker DRX-500, or AVANCE III-600 spectrometers with TMS as an internal standard. Silica gel and Sephadex LH-20 were used for column chromatography (CC). Medium-pressure liquid chromatography (MPLC) was performed on a Lisui EZ Purify III System including pump manager P03, detector modules P02, and fraction collector P01 and columns packed with RP-18 gel. Fractions were monitored by TLC and spots were visualized by heating silica gel plates sprayed with 10% H_2_SO_4_ in EtOH.

### 2.2. Fungus Materials

The fungal strain (PF02-2) was isolated from decaying wood in a freshwater habitat, which was collected in Fangchenggang City, Guangxi Province, China, in May 2016. This strain was reported as a new species viz *Tubeufia rubra* by Lu et al. [29]. The strain is preserved at the Engineering and Research Center for Southwest Bio-Pharmaceutical Resources of National Education Ministry of China, Guizhou University (GZCC 16038) and China Center for Type Culture Collection (CCTCC NO. M2019957), Wuhan University.

### 2.3. Fermentation, Extraction and Isolation

A stock culture of the strain was maintained at 4 °C on PDA medium (Solarbio). The stock culture was inoculated into potato dextrose agar medium (PDA) at 28 °C for 17 days. A flask of 250 mL containing 100 mL of seed medium (PDB) was inoculated with six agar chunks with an area of 1 × 1 cm^2^ from a well-grown culture of PF02-2 and shaken on a rotary shaker at 28 °C and 180 rpm for 3 days according to the preliminary experimental results. The seed cultures were then used to inoculate the production-scale fermentation. Each seed culture (2 mL/piece) was transferred into a flask of 1 L containing 200 g oat and 150 mL distilled water and fermented at 28 °C for 105 days. The whole fermented cultures (30 kg) were extracted three times with EtOAc at room temperature for 48 h on a rotary shaker (50 rpm), and the combined EtOAc extracts (100 L) were concentrated under reduced pressure to give a crude extract (2027.17 g) for further separation.

The crude extract (2027.17 g) was performed with a silica gel column (100–200 mesh, 12 × 150 cm) and eluted with petroleum ether, dichloromethane, ethyl acetate and methanol to obtain fractions Fr. 1→4 according to TLC monitor. Fr. 3 (61.4 g) was subjected to C_18_ reversed-phase silica gel (40–75 μm, 10 × 150 cm) and eluted with a gradient of MeOH/H_2_O (20:80–100:0, *v/v*) to give subfractions Fr. 3-1→Fr. 3-18. Fr. 3-5 (767 mg) was performed with a silica gel column (200–300 mesh, 1.7 × 30.5 cm) and eluted with dichloromethane -methanol-methanoic acid system (200:20:1→200:40:1, *v/v*) to give subfractions Fr. 3-5-1→Fr. 3-5-2. Fr. 3-5-2 (28 mg) was purified by Sephadex LH-20 (20–150 μm, 1.0 × 50 cm) using methanol to obtain **18** (17 mg). Fr. 3-6 (3.6 g) was subjected to a silica gel column (200–300 mesh, 3.2 × 45.7 cm) eluted with dichloromethane -methanol-methanoic acid system (220:20:1→200:40:1, *v/v*) to give five subfractions Fr. 3-6-1→Fr. 3-6-5. Fr. 3-6-2 (22 mg) was purified by Sephadex LH-20 (20–150 μm, 1.0 × 50 cm) using dichloromethane -methanol (1:1, *v/v*) to obtain **15** (9.0 mg). Fr. 3-6-3 (2.6 g) was purified on C_18_ reversed-phase silica gel (40–75 μm, 4.9 × 23 cm) and eluted with a gradient of MeOH/H_2_O (40:60→100:0, *v/v*) to give subfractions Fr. 3-6-3-1→Fr. 3-6-3-4. Fr. 3-6-3-3 (1.8 g) was subjected to a silica gel column (200–300 mesh, 3.2 × 45.7 cm) eluted with dichloromethane -methanol system (15:1→10:1, *v/v*) to yield **9** (12.3 mg). Fr. 3-7 (4.7 g) was performed with C_18_ reversed-phase silica gel (40–75 μm, 4.9 × 23 cm) and eluted with a gradient of MeOH/H_2_O (30:70→100:0, *v/v*) to give subfractions Fr. 3-7-1→ Fr. 3-7-7. Fr. 3-7-5 (213 mg) was purified via a silica gel (200–300 mesh, 1.7 × 23 cm) column eluted with dichloromethane -methanol system (20:1→10:1, *v/v*) to yield **11** (58 mg). Fr. 3-7-6 (298 mg) was subjected to a silica gel column (200–300 mesh, 1.7 × 30.5 cm) eluted with dichloromethane -methanol system (15:1→10:1, *v/v*) to yield **6** (72 mg). Fr. 3-8 (2.6 g) was performed with silica gel column chromatography (CC) (200–300 mesh, 3.2 × 45.7 cm) using CHCl_2_ -CH_3_OH (100:1→20:1, *v/v*) to obtain subfractions Fr. 3-8-1→Fr. 3-8-4. Fr. 3-8-3 (357 mg) was subjected to a silica gel column (200–300 mesh, 1.7 × 30.5 cm) eluted with dichloromethane -methanol system (20:1→10:1, *v/v*) to yield **13** (5.7 mg). Fr. 3-15 (9.23 g) was performed with silica gel CC (200–300 mesh, 8.0 × 61 cm) using CHCl_2_ -CH_3_OH (15:1→5:1, *v/v*) to obtain subfractions Fr. 3-15-1→Fr. 3-15-10. Fr. 3-15-4 (760 mg) was subjected to a silica gel column (200–300 mesh, 2.6 × 30.5 cm) eluted with dichloromethane -methanol system (15:1→10:1, *v/v*) to yield **1** (2.8 mg). Fr. 3-15-5 (358 mg) was subjected to a silica gel column (200–300 mesh, 1.7 × 30.5 cm) eluted with ethyl acetate-methanol system (50:1→30:1, *v/v*) to yield subfractions Fr. 3-15-5-1→3-15-5-4. Fr. 3-15-5-1 (52 mg) was purified by C_18_ reversed-phase silica gel (40–75 μm, 1.5 × 23 cm) and eluted with MeOH/H_2_O (80:20→100:0, *v/v*) to give **5** (27.3 mg). Fr. 3-15-6 (152 mg) was subjected to a silica gel column (200–300 mesh, 1.7 × 23 cm) and eluted with dichloromethane-methanol system (80:1→20:1, *v/v*) to yield **16** (3.1 mg) and fraction Fr. 3-15-6-1. Fr. 3-15-6-1 (109 mg) was purified by a silica gel column (200–300 mesh, 1.7 × 23 cm) and eluted with dichloromethane-methanol system (8:1, *v/v*) to yield **8** (13.6 mg). Fr. 3-15-7 (33 mg) was purified by a silica gel column (200–300 mesh, 1.7 × 23 cm) and eluted with dichloromethane-methanol system (20:1, *v/v*) to yield **14** (16.6 mg). Fr. 3-15-8 (104 mg) was performed with C_18_ reversed-phase silica gel (40–75 μm, 1.7 × 23 cm) and eluted with MeOH/H_2_O (95:25→100:0, *v/v*) to obtain **10** (53 mg). Fr. 3-15-9 (186 mg) was purified by a silica gel column (200–300 mesh, 1.7 × 23 cm) eluted with dichloromethane-methanol system (10:1, *v/v*) to yield **2** (7.3 mg). Fr. 3-16 (367 mg) was purified by a silica gel column (200–300 mesh, 1.7 × 30.5 cm) eluted with dichloromethane-methanol system (30:1→5:1, *v/v*) to obtain subfractions Fr. 3-16-1→Fr. 3-16-3. Fr. 3-16-3 (290 mg) was performed with a silica gel CC (200–300 mesh, 1.7 × 30.5 cm) using CHCl_2_ -CH_3_OH (15:1→5:1, *v/v*) to obtain compound **12** (5.8 mg) and **3** (159 mg).

Fr. 4 (35.77 g) was subjected to C_18_ reversed-phase silica gel (40–75 μm, 10 × 150 cm) and eluted with a gradient of MeOH/H_2_O (20:90→100:0, *v/v*) to give subfractions Fr. 4-1→Fr. 4-17. Fr. 4-11 (8.7 g) was purified on C_18_ reversed-phase silica gel (40–75 μm, 4.9 × 46 cm) and eluted with a gradient of MeOH/H_2_O (30:70→100:0, *v/v*) to give subfractions Fr. 4-11-1→Fr. 4-11-17. Fr. 4-11-6 (123 mg) was performed with a silica gel CC (200–300 mesh, 1.7 × 23 cm) using ethyl acetate-methanol (10:1→5:1, *v/v*) to obtain compound **4** (3.1 mg). Fr. 4-11-12 (2.35 g) was performed with a silica gel CC (200–300 mesh, 2.6 × 30.5 cm) using ethyl acetate-methanol (10:1-5:1, *v/v*) to obtain compound **7** (120 mg) and **17** (324 mg).

### 2.4. Spectral Data

Rubracin D (**1**) Colorless oil; [α${]}_{D}^{28.8}$ = +62.06 (c = 0.13, CH_3_OH); UV (MeOH) λ_max_ (log ε) 196.5 (4.17) and 193 (4.08) nm; IR (KBr) *v*_max_ 3400, 1739, 1644, 1243, 1072 cm^−1^; ^1^H (500 MHz) and ^13^C NMR (125 MHz), see Table 1 (MeOH-*d*_4_); HR-ESI-MS *m/z* 1005.60938 [M + Na]^+^ (calcd for C_53_H_90_O_16_Na, 1005.61211).

Rubracin E (2) Colorless oil; [α${]}_{D}^{28.8}$= +73.58 (c = 0.11, CH_3_OH); UV (MeOH) λ_max_ (log ε) 196 (4.35); IR (KBr) *v*_max_ 3421, 1740, 1630, 1243, 1073 cm^−1^; ^1^H (500 MHz) and ^13^C NMR (125 MHz), see Table 1 (MeOH-*d*_4_); HR-ESI-MS *m/z* 981.61029 [M + Na]^+^ (calcd for C_51_H_90_O_16_Na, 981.61211).

### 2.5. Cytotoxicity Assays

MCF-7/ADM, K562/ADM and A549/ADM cells were purchased from Shanghai MEIXUAN Biological Science and Technology Co., Ltd. (Shanghai, China). Cytotoxicity assays were performed via the CCK-8 method in 96-well microplates [30]. A total of 100 µL cells were cultured at 37 °C under a humidified atmosphere of 5% CO_2_ in RPMI 1640 medium supplemented with 10% fetal bovine serum and dispersed in replicate 96-well plates with 3 × 10^4^cells/well for 24 h. After incubation, MCF-7/ADM cells were treated with 6.25, 12.5, 25, 50, 100, and 200 μg/mL tested compounds (**1**–**2**) or ADM (used as positive control) for 48 h at 37 °C in a humidified atmosphere of 5% CO_2_, while A549/ADM and K562/ADM cells were treated with six concentration gradients tested compounds (**1**–**2**) or ADM (diluted twice at an initial concentration of 25 μg/mL) for 48 h at 37 °C in a humidified atmosphere of 5% CO_2_. At the end of incubation, each well was added to 10 CCK-8 solution and continued to incubate for 3 h at 37 °C in a humidified atmosphere of 5% CO_2_. Finally, OD was measured at wavelength 490 nm. A total of 10 µL DMSO and 90 µL medium suspension were used as negative control and each assay was performed in triplicate. 

### 2.6. Reversed MDR Assays

Reversed MDR assays were also performed using the CCK-8 method and its experimental procedure was the same as that of cytotoxicity assays. The difference mainly lay in the treatment of cells with tested compounds (**1**–**2**). In the assay on MCF-7/ADM, cells were treated with ADM at different concentrations (6.25, 12.5, 25, 50, 100, and 200 μg/mL) combined with 5, 10, 20 μg/mL compounds (**1**–**2**) and Verapamil (used as positive control), respectively. In the assays on A549/ADM and K562/ADM, cells were treated with ADM at different concentrations (0.78125, 1.5625, 3.125, 6.25, 12.5, and 25 μg/mL) combined with 5, 10, 20 μg/mL compounds (**1**–**2**) and Verapamil, respectively. Six concentration gradients ADM (6.25, 12.5, 25, 50, 100, or 200 μg/mL) were used as a negative control.

### 2.7. Western Blot (WB) Assay

MCF-7/ADM cells were seeded at a density of 2 × 10^5^ cells/well in 6-well plates and incubated for 24 h. Each tumor cell was exposed to each test compound at 0, 5, 10, and 20 μg/mL combined with ADM at the concentration of 65 μg/mL in triplicate for 48 h. Cells were washed three times with PBS and lysed directly in the RIPA (Radio Immunoprecipitation Assay) buffer. The lysates were centrifuged at 12,000 rpm for 5 min, and the supernatants were collected. The protein concentrations of the supernatants were determined using a BCA protein assay kit. SDS-PAGE was used to separate whole cell lysates having equivalent amounts of protein to be transferred to polyvinylidene difluoride (PVDF) membranes. PVDF membranes were incubated with primary antibodies and then with HRP-conjugated secondary antibodies. The proteins were then detected using the ECL detection kit.

### 2.8. Data Analysis

Each experiment was performed at least in triplicate. The IC_50_ values were calculated by Modified Kou’s method. The grayscale values were calculated by IPP 6.0. One-way analysis of variance (one-way ANOVA) was used to evaluate the data with the following significance levels: * *p* < 0.05, ** *p* < 0.01, *** *p* < 0.001, and *p* < 0.0001. The results are expressed as the mean standard deviation (SD) and activity-related figures were calculated by GraphPad Prism 8.0.

## 3. Results

### 3.1. Structure Identification of Compounds **1–18**

Compound (**1**), namely Rubracin D, was obtained as colorless oil; The structure of compound **1** was shown in Figure 1; [*α*${]}_{D}^{28.8}$ = +62.06 (*c* 0.13, MeOH) (Appendix A); Its molecular formula was established as C_53_H_90_O_16_ by HR-ESI-MS ([M + Na]^+^ at m/z 1005.60938) (Appendix A) indicating 9 degrees of unsaturation. The UV (Appendix A) absorption maxima of **1** at 196 nm indicated the presence of double bonds, while the IR (Appendix A) absorption bands at 3400 and 1740 cm^−1^ revealed the presence of hydroxy and carbonyl functionalities. The ^1^H NMR (Appendix A), ^13^C NMR and DEPT spectra (Appendix A) and combined with HSQC spectra (Appendix A) of **1** displayed the presence of sugar and unsaturated long-chain fatty acid ester moieties. the 1D NMR data of compound **1** was shown in Table 1. The ^1^H-^1^H COSY (Appendix A) experiment revealed that there are some fragments H-1→H-2→H-3, H-1′→H-2′→H-3′→H-4′→H-5′→H-6′, H-1″→H-2″→H-3″→H-4″→H-5″→H-6″, H-2″′→H-3″′, H-8″′→H-9″′→H-10″′→H-11″′→H-12″′→H-13″′→H-14″′, H-3″″→H-4″″, and H-8″″→H-9″″→H-10″″→H-11″″→H-12″″→H-13″″→H-14″″(see Figure 2). Analysis of the HMBC spectrum (Appendix A), one group of signals at *δ*_C_ 105.4, 72.3, 74.7, 70.1, 74.8, 68.2 and *δ*_H_ 4.24 (1H, d, *J* = 7.4 Hz) revealed the presence of a 6-*O*-substituted *β*-*D*-galactopyranose unit (Appendix A). Another group of signals at *δ*_C_ 100.5, 70.2, 70.9, 71.2, 70.0, 65.0 and *δ*_H_ 4.85 (1H, br s) indicated the presence of a *α*-*D*-galactopyranose unit. Moreover, the other group of signals at *δ*_C_ 64.0, 68.7, 71.7 and *δ*_H_ 4.22 (1H, dd, *J* = 12.0, 6.6 Hz), 4.44 (1H, dd, *J* = 12.1, 2.9 Hz), 3.73 (1H, Overlap), 3.91 (1H, dd, *J* = 10.9, 5.4 Hz), and 5.25 (1H, m) revealed a glycerol moiety. Meanwhile, the anomeric H-1′ of the *β*-*D*-galactopyranosyl at *δ*_H_ 4.24 was correlated with *δ*_C_ 68.7, and 72.3 implied the anomeric H-1′ of the *β*-*D*-galactopyranosyl connected with the C-3 of the glycerol moiety. Another anomeric H-1″ of the *α*-*D*-galactopyranosyl was correlated to the C-6′ of the *β*-*D*-alactopyranosyl, which was supported by the correlations of the proton at *δ*_H_ 4.85 to *δ*_C_ 68.2, 70.9, and 71.2 (Figure 3). All of the above findings revealed that compound **1** was a 1, 2-di-*O*-acyl-3-*O*-[*α*-*D*-galactopyranosyl-(1″→6′)-*O*-*β*-*D*-galactopyranosyl]-glycerol. Furthermore, the ^13^C NMR spectrum of **1** showed two carbonyl carbon signals at *δ*_C_ 175.0 (C-1′′′) and 174.6 (C-1), and their locations were confirmed by the correlations of H-1 with C-1′′′ and H-2 with C-1′′′′ detected in the HMBC experiment. In the ^1^H and ^13^C NMR spectra, the unsaturated long-chain fatty acid ester signals at [14.5 (q), 0.90 (6H, d, *J* = 6.9 Hz)], [26.6 (t), 2.77 (4H, t, *J* = 6.6 Hz)], [2.82 (t), 2.05 (8H, m)], and [129.1–131.0 (d), 5.30–5.36 (8H, m)] suggested that compound **1** has two linolenoyl units, which was further verified by the molecular formula of **1**. The detailed analysis of 1D NMR and 2D NMR, of the chemical structure of compound **1** were very similar to those of 1, 2-*O*-(9*Z*, 12*Z*-octadecadienoyl)-3-*O*-[α-*D*-galactopyranosyl-(1″→6′)-*O*-*β*-*D*-galactopyranosyl]-glycerol [31], except that a hydrogen atom was replaced by acetyl group in compound 1, 2-*O*-(9*Z*, 12*Z*-octadecadienoyl)-3-*O*-[*α*-*D*-galactopyranosyl-(1″→6′)-*O*-*β*-*D*-galactopyranosyl]-glycerol. The acetyl group connected with C-6″ was supported by the correlations of H-6″ with 172.7, 71.2, and 70.9 in the HMBC experiment. Except for *β*-*D*-galactopyranose and *α*-*D*-galactopyranose, the C-2 configuration of compound **1** was determined to be *S* by comparison with the optical rotation values and the chemical shift of C-2 reported in the literature [32]. As a result, the structure of **1** was determined as (2*S*)-1, 2-*O*-linoleoyl-3-*O*-[α-D-galactopyranosyl-(1″→6′)- (6″-*O*-acetyl)-*O*-*β*-*D*-galactopyranosyl]-glycerol.

Compound (**2**), [*α*${]}_{D}^{28.8}$ = +73.85 (*c* 0.11, MeOH) (Appendix A), was obtained as a colorless oil and possessed a molecular formula of C_51_H_90_O_16_, established by HR-ESI-MS ([M + Na]^+^ at m/z 981.61029) (Appendix A). The UV (Appendix A) absorption maxima of 2 at 196 nm indicated the presence of double bands, while the IR spectra (Appendix A) revealed the presence of hydroxy and carbonyl functionalities; the chemical structure of **2** was displayed in Figure 1. In an analysis of ^1^H NMR (Appendix A), ^13^C NMR, and DEPT spectra (Appendix A), the 1D NMR data (see Table 1) were very similar to those of **1** except for the difference of another long-chain saturated fatty acid ester unit, This assumption was supported by the signals at *δ*_C_ 175.0, 35.1, 32.7, 30.2–30.8, 26.0, 23.7, 14.5 and *δ*_H_ 2.32 (2H, d, *J* = 7.3 Hz), 1.60 (2H, m), 0.90 (3H, d, *J* = 6.9 Hz). The detailed analysis of HSQC (Appendix A), ^1^H-^1^H COSY (Appendix A), and HMBC (Appendix A) correlations together with the molecular formula confirmed that compound **2** was (2*S*)- 1-*O*-palmitoyl-2-*O*-linoleoyl-3-*O*-[*α*-*D*-galactopyranosyl-(1′′→6′)- (6″-*O*-acetyl)-*O*-*β*-*D*-galactopyranosyl]-glycerol, and namely Rubracin E. The key ^1^H-^1^H COSY and HMBC correlations of compound **2** were shown in Figure 3.

Furthermore, sixteen known compounds were also isolated. According to the comparison with the reported NMR data and mass data, sixteen compounds [Figure 1] were identified as 1-*O*-palmitoyl-3-*O*-[*α*-*D*-galactopyranosyl-(1→6)-*β*-*D*-galactopyranosyl]- glycerol (**3**) [33], gingerglycolipid C (**4**) [34], gingerglycolipid B (**5**) [35], 3-*O*-octadeca-9*Z*,12*Z*,15*Z*-trienoylglyceryl-6′-*O*-(*α*-*D*-galactopyranosyl)-*β*-*D*-galactopyranoside (**6**) [35], 1-*O*-palmitoyl-2-*O*-linoleoyl-3-*O*-[*α*-*D*-galactopyranosyl-(1→6)-*β*-*D*-galactopyranosyl]-glycerol (**7**) [36], 1-*O*-octadecanoyl-2-*O*-(9*Z*,12*Z*-octadecadienoyl)-3-*O*-[*α*-*D*-galactopyranosyl-(1′′→6′)-*O*-β-*D*-galactopyranosyl] glycerol (**8**) [37], 1-*O*-linoleoyl-2-*O*-oleoyl-3-*O*-[*α*-*D*-galactopyranosyl-(1→6)-*β*-*D*-galactopyranosyl]-glycerol (**9**) [38], 2,3-*O*-dioctadeca-9*Z*,12*Z*-dienoylglyceryl-6′-*O*-(*α*-*D*-galactopyranosyl)-*β*-*D*-galactopyranoside (**10**) [35], 2-*O*-octadeca-9*Z*,12*Z*-dienoyl-9*Z*,12*Z*,15*Z*-trienoylglyceryl-6′-*O*-(*α*-*D*-galactopyranosyl)-*β*-*D*-galactopyranoside (**11**) [39], 1-*O*-(9*Z*,12*Z*-octadecadienoly)-3-*O*-*β*-galactopyranosylglycerol (**12**) [40], 3-*O*-octadeca-9*Z*,12*Z*,15*Z*-trienoylglyceryl-*O*-*β*-*D*-galactopyranoside (**13**) [33], 1-*O*-oleoyl-2-*O*-myristoyl-glyceryl-*O*-*β*-*D*-galactopyranoside (**14**) [35], 1,2-*O*-diacyl-3-*O*-*β*-*D*-galactopyranosyl glycerols (**15**) [37], 2,3-*O*-dioctadeca-9*Z*,12*Z*-dienoylglyceryl-*O*-*β*-*D*-galactopyranoside (**16**) [35], 1-*O*-(9*Z*, 12*Z*-octadecadienoyl)-3-*O*-[*β*-*D*-galactopyranosyl-(1→6)-*O*-*β*-*D*-galactopyranosyl-(1→6)-*O*-*β*-*D*-galactopyranosyl] glycerol (**17**) [41], 1,2-*O*-(9*Z*,12*Z*-octadecadienoyl)-3-*O*-[*α*-*D*-galactopyranosyl-(1′′′′′→6′′′′)-*O*-*β*-*D*-galactopyranosyl-(1′′′′→6′′′)-*O*-*β*-*D*-galactopyranosyl]-glycerol (**18**) [31]. Their chemical structures were described in Figure 4.

### 3.2. Results of Bioactivity Assays

#### 3.2.1. Cytotoxicity Evaluation In Vitro

The new compounds **1** (PF70) and **2** (PF52) were subjected to cytotoxicity evaluation on the three human cancer cell lines viz MCF-7/ADM, K562/ADM and A549/ADM using the CCK-8 method. Both compounds **1** and **2** showed cytotoxic activity on the cell lines with an inhibition ratio of less than 20% at a concentration lower than 50 μg/mL. The results indicated that both compounds were applicable for studying MDR reversal activity.

#### 3.2.2. MDR Reversal Effects of Compound **1** and **2** In Vitro

The MDR reversal assay was performed using the anticancer drug adriamycin (ADM) at different concentrations combined with the test compounds at concentrations of 0, 5, 10 and 20 μg/mL, respectively, and Vrp as the positive control. The reversal activities of compound **1** and compound **2** against the three tumor cells were shown in Figure 5 and Table 2. In the assay on MCF-7/ADM, both compounds **1** (See Table 2 and Figure 5A) and **2** (See Table 2 and Figure 5B) showed certain reversal effects, but not as effective as those of Vrp. While in the assay on K562/ADM, both compounds **1** (See Table 2 and Figure 5C) and **2** (See Table 2 and Figure 5D) were proved to have significant MDR reversal effects with IC_50_ values range from 3.813 ± 0.123 to 2.360 ± 0.208 μg/mL and 3.764 ± 0.297 to 2.448 ± 0.225 μg/mL, respectively, which were higher than that of Vrp with an IC_50_ value range from 4.038 ± 0.514 to 2.554 ± 0197 μg/mL at 5, 10, 20 μg/mL, respectively. However, one-way variance contrast analysis (ANOVA) showed no statistical difference (*p* > 0.05). In the assay on A549/ADM, compound **1** (See Table 2 and Figure 5E) displayed significant MDR reversal effects with IC_50_ value range from 2.897 ± 0.072 to 1.841 ± 0.079 μg/mL, which were higher than those of Vrp at 5 and 20 μg/mL, respectively, with IC_50_ value ranges from 2.927 ± 0.226 to 1.848 ± 0.015 μg/mL. While the results of the ANOVA indicated no statistical difference (*p* > 0.05). Compound **2** (See Table 2 and Figure 5F) also showed certain reversal effects, but not as effective as those of Vrp. Meanwhile, as also shown in Figure 6 and Figure 7, the inhibition ratio curves of compound **1** and **2** combined with ADM on the three drug-resistant cell lines indicated that both compounds significantly increased the ADM sensitivities of ADM-resistant cells with significant dose-dependence.

Finally, we tested the P-gp expressions in MCF-7/ADM cells with the treatment of ADM at the concentration of 65 μg/mL combined with compound **1** and **2** at 0, 5, 10, 20 μg/mL, respectively, for 48 h. Our Western blot assays showed that the test with compound **1** at the concentration of 20 μg/mL had a significant suppression effect on the expression level of P-gp (Figure 6D), which proved that compound **1** reversed the MDR of the MCF-7/ADM cell line via hindering the overexpression of P-gp. The Western blot assays also showed that the test of compound **2** at the concentrations of 10 and 20 μg/mL, respectively, had remarkable suppression effects on the expression level of P-gp (Figure 7D) and therefore reversed the MDR of the MCF-7/ADM cell line.

## 4. Discussion

The strain *Tubeufia rubra* (PF02-2) was first isolated and identified by Dr. Lu of our research group, so there is no relevant report on the metabolites of the strain. In the paper, two new compounds, viz Rubracin D (**1**) and E (**2**), as well as sixteen known constituents (3–18) were isolated from the crude ethyl acetate extract of the fermentation broth of the fungal strain *Tubeufia rubra* (PF02-2). Their chemical structures were identified on the basis of their spectroscopic spectra data, as well as via comparison with literature data. Compounds (**1**–**18**) were isolated from the fungal genus *Tubeufia* for the first time. Bioassay results indicated that the new compounds (**1**, **2**) showed significant MDR reversal activities on MCF-7/ADM, K562/ADM and A549/ADM cell lines. It is worth noting that both compounds (**1**, **2**) were proved to have significant MDR reversal effects mostly higher than those of the positive control viz Vrp in the assay on the K562/ADM cell line. Additionally, compound 1 displayed significant MDR reversal effects mostly higher than those of Vrp in the assay on A549/ADM cell line. The results of the Western blot assays proved that both compounds **1** and **2** significantly reversed the MDR of the MCF-7/ADM cell line via hindering the overexpression of P-gp. We, therefore, speculate that the MDR reversal mechanism of both compounds on the other two resistant cancer cell lines viz K562/ADM and A549/ADM was also due to inhibiting the overexpression of P-gp.

## Figures and Tables

**Figure 1 jof-09-00309-f001:**
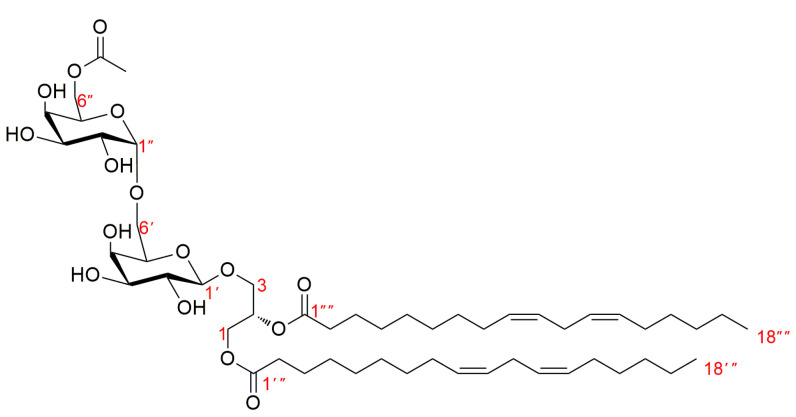
Chemical structure of compound **1** from *Tubeufia rubra* PF02-2.

**Figure 2 jof-09-00309-f002:**
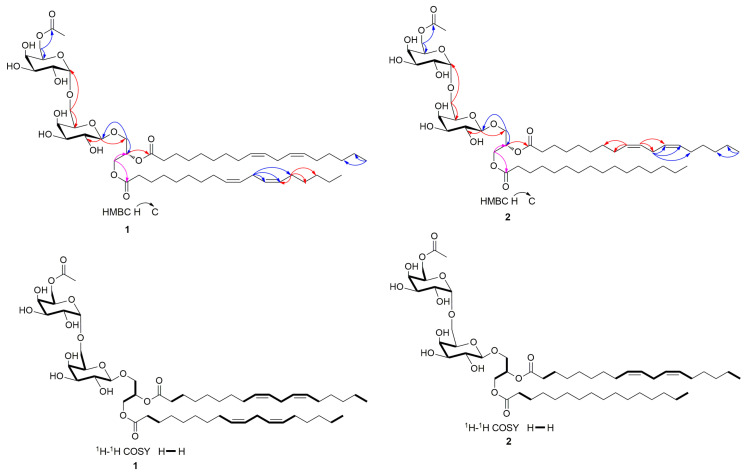
Key ^1^H-^1^H COSY and HMBC correlations of compounds **1** and **2**.

**Figure 3 jof-09-00309-f003:**
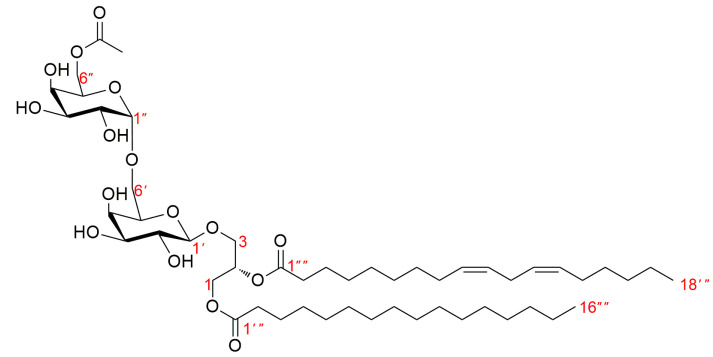
Chemical structures of compound **2** from *Tubeufia rubra* PF02-2.

**Figure 4 jof-09-00309-f004:**
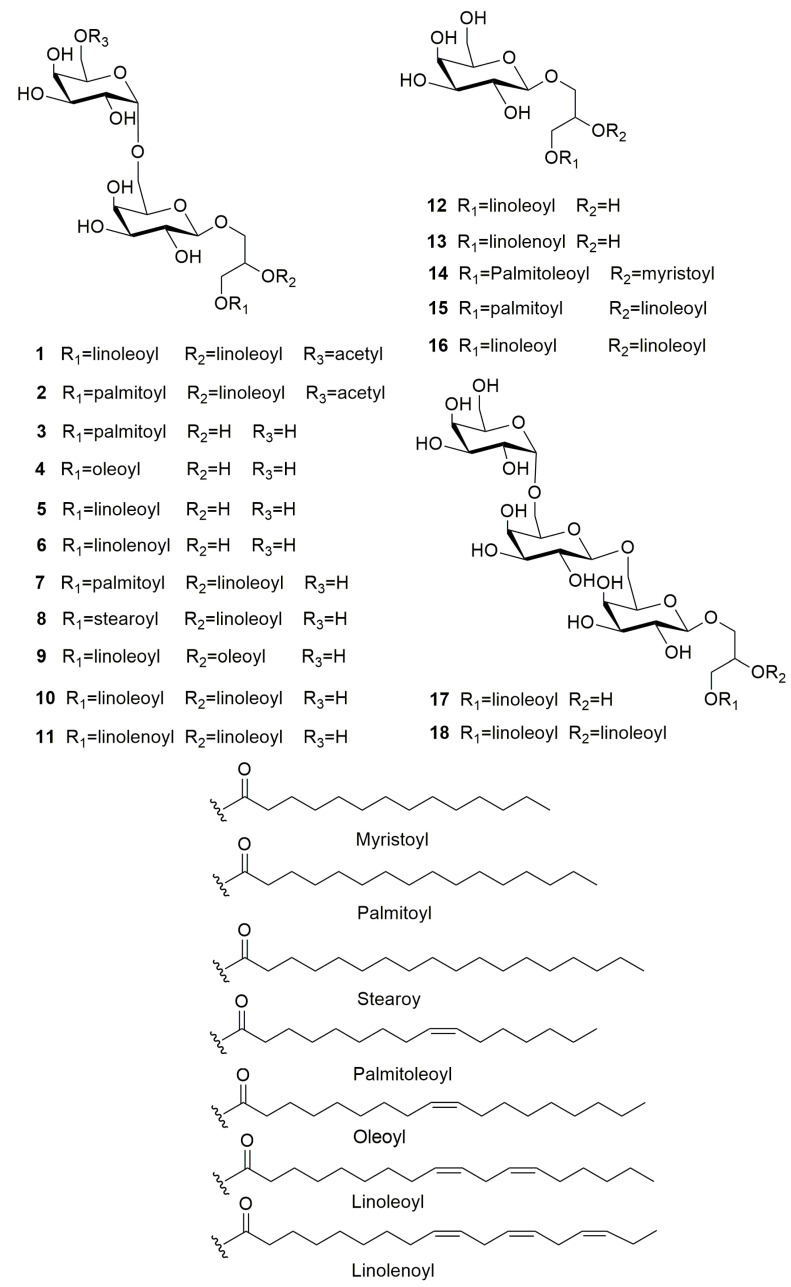
Chemical structures of compounds **1**–**18** from *Tubeufia rubra* PF02-2.

**Figure 5 jof-09-00309-f005:**
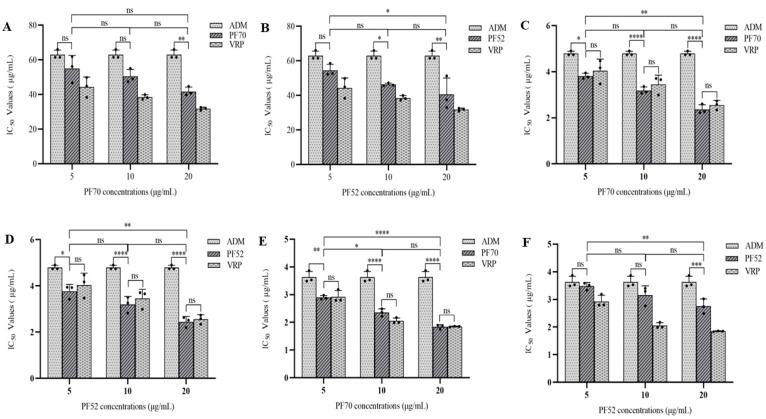
IC_50_ values of ADM at different concentrations combined with compound **1** (PF70) and **2** (PF52) at concentrations of 0, 5, 10, 20 μg/mL, respectively: (**A**) compound **1** (PF70) against MCF-7/ADM; (**B**) compound **2** (PF52) against MCF-7/ADM; (**C**) compound **1** (PF70) against K562/ADM; (**D**) compound **2** (PF52) against K562/ADM; (**E**) compound **1** (PF70) against A549/ADM; (**F**) compound **2** (PF52) against A549/ADM. All experiments were performed in triplicate. ns *p* > 0.05, * *p* < 0.05, ** *p* < 0.01, and *** *p* < 0.001, and **** *p* < 0.0001.

**Figure 6 jof-09-00309-f006:**
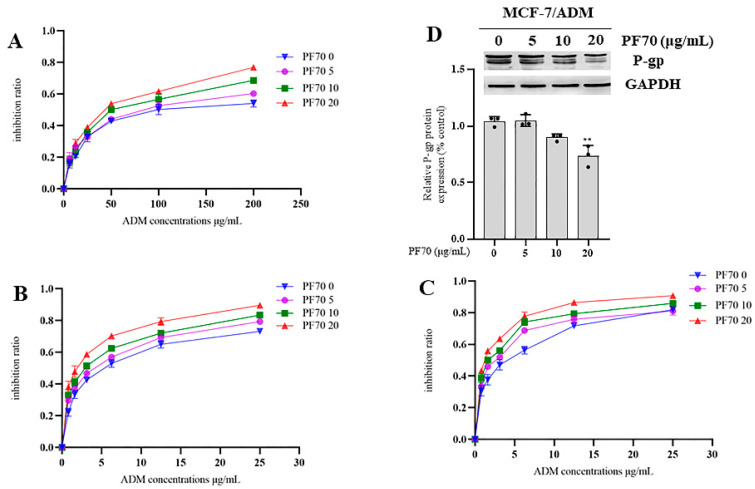
Inhibition ratios of ADM at different concentrations combined with compound **1** (PF70) at concentrations of 0, 5, 10 and 20 μg/mL, respectively: (**A**) MCF-7/ADM; (**B**) K562/ADM; (**C**) A549/ADM; (**D**) Effect of **1** (PF 70) on P-gp expression. All experiments were performed in triplicate. ** *p* < 0.01.

**Figure 7 jof-09-00309-f007:**
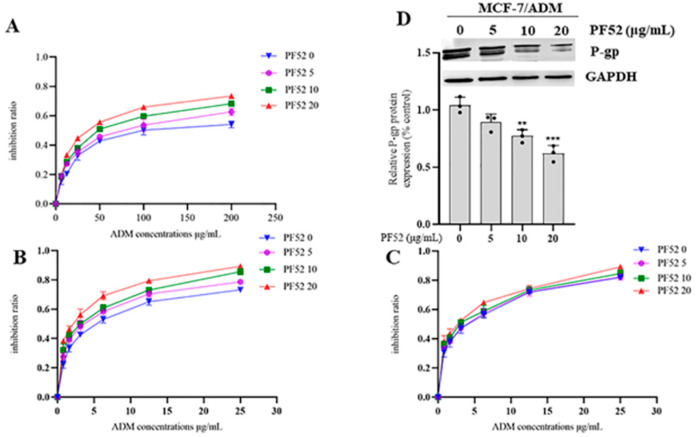
Inhibition ratios of ADM at different concentrations combined with compound 2 (PF52) at concentrations of 0, 5, 10 and 20 μg/mL, respectively: (**A**) MCF-7/ADM; (**B**) K562/ADM; (**C**) A549/ADM; (**D**) Effect of 2 (PF 52) on P-gp expression. All experiments were performed in triplicate. ** *p* < 0.01, and *** *p* < 0.001.

**Table 1 jof-09-00309-t001:** 1D NMR data of Rubracin D (**1**) and E (**2**) in MeOD (^1^H-NMR at 500 MHz; ^13^C-NMR at 125 MHz).

Position	1		Position	2	
	*δ* _C_	*δ*_H_, mult (*J* in Hz)		*δ* _C_	*δ*_H_, mult (*J* in Hz)
1	64.0, CH_2_	4.44, dd, (12.1, 2.9) 4.22, dd, (12.0, 6.6)	1	63.9, CH_2_	4.44, dd, (12.0, 2.9) 4.22, overlap
2	71.7, CH	5.25, m	2	71.7, CH	5.24, m
3	68.7, CH_2_	3.91, dd, (10.9, 5.4) 3.74, overlap	3	68.7, CH_2_	3.91, dd, (10.9, 5.4) 3.74, overlap
1′	105.4, CH	4.24, d, (7.4)	1′	105.4, CH	4.24, d, (7.3)
2′	72.3, CH	3.51, dd, (9.7, 7.2)	2′	72.3, CH	3.51, dd, (9.7, 7.3)
3′	74.7, CH	3.47, dd, (9.7, 3.2)	3′	74.7, CH	3.47, dd, (9.7, 3.2)
4′	70.1, CH	3.85, dd, (3.1, 0.6)	4′	70.1, CH	3.85, dd, (3.1, 0.6)
5′	74.8, CH	3.70, overlap	5′	74.8, CH	3.70, overlap
6′	68.2, CH_2_	3.89, overlap 3.64, dd, (10.0, 5.5)	6′	68.2, CH_2_	3.89, overlap 3.64, dd, (10.0, 5.5)
1″	100.5, CH	4.85, br s	1″	100.5, CH	4.85, br s
2″	70.2, CH	3.76, overlap	2″	70.2, CH	3.76, overlap
3″	70.9, CH	3.72, overlap	3″	70.9, CH	3.72, overlap
4″	71.2, CH	3.87, overlap	4″	71.2, CH	3.87, overlap
5″	70.0, CH	4.04, br t, (6.7)	5″	70.0, CH	4.04, br t, (6.7)
6″	65.0, CH_2_	4.20, d (6.4)	6″	65.0, CH_2_	4.20, d (6.4)
1′″″	172.7, C		1′″″	172.7, C	
2′″″	20.9, CH_3_	2.06, s	2′″″	20.9, CH_3_	2.06, s
1′″	175.0, C		1′″	175.0, C	
2′″	35.1, CH_2_	2.32, t, (7.3)	2′″	35.1, CH_2_	2.32, t, (7.3)
3′″, 3″″	26.0, CH_2_	1.60, m	3′″, 3″″	26.0, CH_2_	1.60, m
4′″, 4″″-7′″, 7″″	30.2–30.8, CH_2_	1.28–1.36, m	4′″-13′″	30.2–30.8, CH_2_	1.28–1.36, m
8′″, 8″″, 14′″, 14″″	28.2, CH_2_	2.05, m	14′″, 16″″	32.7, CH_2_	1.28–1.36, m
9′″, 9″″	131.0 ^a^, CH	5.34, m	15′″, 17″″	23.6, CH_2_	1.28–1.36, m
10′″, 10″″	129.2 ^b^, CH	5.32, m	16′″, 18″″	14.5, CH_3_	0.90, t, (6.9)
11′″, 11″″	26.6, CH_2_	2.77, t, (6.6)	1″″	174.6, C	
12′″, 12″″	129.1 ^b^, CH	5.32, m	2″″	35.0, CH_2_	2.31, t, (7.3)
13′″, 13″″	130.9 ^a^, CH	5.34, m	4″″-7″″	30.2–30.8, CH_2_	1.28–1.36, m
15′″, 15″″	30.2–30.8, CH_2_	1.28–1.36, m	8″″, 14″″	28.2, CH_2_	2.05, m
16′″, 16″″	32.7, CH_2_	1.28–1.36, m	9″″	131.0 ^a^, CH	5.34, m
17′″, 17″″	23.7, CH_2_	1.28–1.36, m	10″″	129.1 ^b^, CH	5.32, m
18′″, 18″″	14.5, CH_3_	0.90, t, (6.9)	11″″	26.6, CH^2^	2.77, t, (6.4)
1′″″	174.6, C		12″″	129.0 ^b^, CH	5.32, m
2′″″	35.0, CH_2_	2.31, t, (7.3)	13″″	130.8 ^a^, CH	5.34, m
			15″″	30.2–30.8, CH_2_	1.28–1.36, m

Assignments with the same superscript (a, b) and in the same column may be interchangeable.

**Table 2 jof-09-00309-t002:** The reversal effects of compounds toward MCF-7/ADM, K562/ADM and A549/ADM.

Samples		IC_50_ (μg/mL)		
		MCF-7/ADM	K562/ADM	A549/ADM
**1**	5 μg/mL	54.958 ± 7.595	3.813 ± 0.123	2.897 ± 0.072
	10 μg/mL	50.456 ± 4.071	3.182 ± 0.157	2.358 ± 0.133
	20 μg/mL	41.726 ± 2.585	2.360 ± 0.208	1.841 ± 0.079
**2**	5 μg/mL	54.487 ± 3.538	3.764 ± 0.297	3.480 ± 0.129
	10 μg/mL	46.289 ± 0.825	3.195 ± 0.347	3.155 ± 0.338
	20 μg/mL	40.567 ± 9.532	2.448 ± 0.225	2.755 ± 0.263
Vrp	5 μg/mL	44.371 ± 5.746	4.038 ± 0.514	2.927 ± 0.226
	10 μg/mL	38.471 ± 1.397	3.452 ± 0.401	2.051 ± 0.107
	20 μg/mL	31.799 ± 0.875	2.554 ± 0.197	1.848 ± 0.015
ADM		63.031 ± 2.616	4.797 ± 0.093	3.630 ± 0.207

All experiments were performed in triplicate. Each value represents the mean ± SD (n = 3). n represents number of experiments.

## Data Availability

Data supporting the reported results are provided in Appendix A. The data from manuscript and Appendix A are available for publication, citation, and use.

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
