# Peer review of "A Saprophytic Fungus Tubeufia rubra Produces Novel Rubracin D and E Reversing Multidrug Resistance in Cancer Cells"

_jof, 2023, doi:10.3390/jof9030309_

Round 1

Reviewer 1 Report

This study revealed 18 chemicals including two new ones from a saprophytic fungus Tubeufia rubra. The new compounds were evaluated for the reversal activity of multidrug resistance (MDR) on MCF-17 7/ADM, K562/ADM and A549/ADM cell lines. And both compound 1 and 2 reversed MDR in the 18 three resistant cancer cell lines with concentration dependence. The structures were established well by the extensive spectroscopic analyses, and the manuscript was well prepared by following the guidelines of Journal of Fungi. Overall, I suggest for publication. Several minor revisions are given below.

1)      Please check the paper, and improve the Language.

2)      Figure 2, the 1H-1H COSY correlations should be demonstrated in the planar structures while the stereochemistry of chiral centers was suggested to be deleted.

3)      Line 13, “sixteen known (3-18) Glyceroglycolipids” should correct to “sixteen known Glyceroglycolipids (3-18)”, and the Numbers of compounds should been bold in all the text.

4)       “compound 1 and 2” should correct to “compounds 1 and 2”, “compound” should been plural in all the text, like lines 18, 252, 255, 259. The “Rubracin D (1) and E (2)” has the same questions.

5)      Line 26, “reverse” should correct to “and reversed”.

6)      In “2.3. Fermentation, Extraction and Isolation”, the sign of the volume ratio was appeared at irregular intervals. The authors should suggest to use the unified Symbol format to describe the solvent ratio changes. And the “v/v” is not italic.

7)      In “2.3. Fermentation, Extraction and Isolation”, both CHCl2 and choroform are used, Please confirm with the author.

8)      Line 164, “3.1. Structrue identification of 1-18” should correct to “3.1. Structrue identification of 1-2”. In the paper, only two new compounds were identified, the known were not identified.

9)      Lines 169-164, “Figure S1” and “Figure S2” correct to “Fig. S1” and “Fig. S2”. “See Fig.S5”, “See Fig.S6”, and “See Fig.S7” delete the “see”.

Author Response

This study revealed 18 chemicals including two new ones from a saprophytic fungus Tubeufia rubra. The new compounds were evaluated for the reversal activity of multidrug resistance (MDR) on MCF-17 7/ADM, K562/ADM and A549/ADM cell lines. And both compound 1 and 2 reversed MDR in the 18 three resistant cancer cell lines with concentration dependence. The structures were established well by the extensive spectroscopic analyses, and the manuscript was well prepared by following the guidelines of Journal of Fungi. Overall, I suggest for publication. Several minor revisions are given below.

1)      Please check the paper, and improve the Language.

Answer: thank you very much for your careful comments, we have revised in our manuscript.

2)      Figure 2, the 1H-1H COSY correlations should be demonstrated in the planar structures while the stereochemistry of chiral centers was suggested to be deleted.

Answer: thank you very much for your useful comments, we have revised it in Figure 2 of the manuscript.

3)      Line 13, “sixteen known (3-18) Glyceroglycolipids” should correct to “sixteen known Glyceroglycolipids (3-18)”, and the Numbers of compounds should been bold in all the text.

Answer: thank you very much for your useful comments, we have revised it in line 13 of the manuscript.

4)      “compound 1 and 2” should correct to “compounds 1 and 2”, “compound” should been plural in all the text, like lines 18, 252, 255, 259. The “Rubracin D (1) and E (2)” has the same questions.

Answer: thank you very much for your careful comments, we have revised in our manuscript.

5)      Line 26, “reverse” should correct to “and reversed”.

Answer: thank you very much for your elaborative comments, we have revised it in line 26 of our manuscript.

6)      In “2.3. Fermentation, Extraction and Isolation”, the sign of the volume ratio was appeared at irregular intervals. The authors should suggest to use the unified Symbol format to describe the solvent ratio changes. And the “v/v” is not italic.

Answer: thank you very much for your elaborative comments, we have revised them in the section 2.3 of our manuscript.

7)      In “2.3. Fermentation, Extraction and Isolation”, both CHCl2 and choroform are used, Please confirm with the author.

Answer: thank you for your thoughtful comment, we have rechecked in section 2.3 and found that the purification and isolation of 18 compounds, only dichloromethane was used, and no chloroform was used. This is our mistake and we have revised in the manuscript.

8)      Line 164, “3.1. Structrue identification of 1-18” should correct to “3.1. Structrue identification of 1-2”. In the paper, only two new compounds were identified, the known were not identified.

Answer: thank you for your careful comments, 1D NMR and HR-ESI-MS data of compounds (318) was added to supplementary material (see Table S1).

9)      Lines 169-164, “Figure S1” and “Figure S2” correct to “Fig. S1” and “Fig. S2”. “See Fig.S5”, “See Fig.S6”, and “See Fig.S7” delete the “see”.

Answer: thank you very much for your careful comments, we have revised in our manuscript.

Reviewer 2 Report

Regarding the manuscript entitled ¨A Saprophytic Fungus Tubeufia rubra Produces Novel Rubracin D and E Reversing Multidrug Resistance in Cancer Cells”. Authors had conducted a chemical study including purification and chemical elucidation of 18 compounds from “whole fermented cultures” of Tubeufia rubra using the fungal strain (PF02-2) growth in PDA and PDB medium and 200 g of oat.  The authors did not specify the density of strain or how they control the content for fermentation. How the authors know that the compounds found are from the fungus and not from the oat used?

Authors had reported 18 glyceroglycolipids and had name them rubracins as the species name T. rubra, however, other glyceroglycolipids had been reported from Avena sativa (oat) [Chemistry and Physics of Lipids 78 (1995) 97; Lipids 36, (2001) 153; Lipids 33, (1998) 355] and authors used 200 g of oat in 450 ml of total volume for fermentation. With this unique extract there is no form to be certain that the compounds are from T. rubra. Authors need more experimental work to define the origin of the compounds. There is only one previous work publishing “compounds from T. rubra” [Rec. Nat. Prod. 16:6 (2022) 622], the paper belongs to the same group and they had the same mistake (The authors cannot affirm with certainty that the metabolites isolated come from the fungus). This paper is not included in the references.

Structures of compounds 1 and 2 can not be completely verified without the appropriate expansions of 1H NMR to verify multiplicity.

Next comments should be attended too.

Line 13. Please write with bold letters all numbers that correspond to compounds.

Line 41. Please write all names of any species with italics.

Section 2.3. Please indicate for each chromatographic process described which kind of column were used, for example, for column chromatography and medium pressure chromatography indicate the column diameter and silica gel size or mesh, if HPLC indicate the column dimensions (long, id and particle size) and pump conditions (flow, temperature, injection vol, describe gradient if used, etc).

Line 82. For convention, when work with reversed phase water is written first and then the organic solvent, you should follow the convention.

Lines 89, 116 and the complete manuscript. As convention, all numbers indicating number of atoms are written with subscript, please follow the convention as least for substances (line 116), for C18 it could be accepted with or without subscript, but be homogeneous, you are using two stiles (see C18 in lines 81, 89, 92). Check in the complete manuscript.

Lines 89-91. Please check the correct amount of each fraction in all experimental section. Authors indicated 2.6 mg for fraction 3-6-3, but 1.8 g for its subfraction 3-6-3-4.

Lines 128 and 132. m/z must be italics.

Lines 126, 128, 130, 132, 141 (cells/mL). There are spaces not needed. Please check the complete manuscript.

Section 2.5. Please indicate the volume and solvent/medium used for compounds tested and positive controls, as well as the concentrations used.

Line 157. Please check writing.

Lines 158, Table 2 and Fig 5. The IC50 values are a measure of the potency of compounds in specific assays models. The most common way to present IC50 is the calculation using Hill equation. I can only find information for Bliss method used for synergy, however, there are no discussion regarding synergy in the manuscript. Could the authors include a reference for the Bliss method used? Could the authors provide an explanation or discussion only for reviewing purposes the reason of showing IC50 values for each concentration assayed? Is that possible to report?  In my opinion it would be more useful if authors present the IC50 values as a single value calculated from different concentrations assayed. For that calculation at least 5 concentrations are needed, and a sigmoid curve obtained. If the authors can't do the math because of the curve shape or concentrations missing, I suggest presenting only as inhibition % with the error measure you choose (std dev, etc).

Line 161. “Results are expressed as the mean standard deviation” or was expressed as the mean ± standard deviation?

Fig 1 and the complete manuscript. Authors are using common names for fatty acid derivatives; it is suggested to use common name for octadecanoic acid too which is stearic acid.

Did the authors follow a specific order to number the compounds?

Authors can improve Fig 1 following the next suggestions according to general conventions: 1) Order the substituents, first the smaller chains, first the saturated chains, thus myristic, palmitic, stearic, oleic, linoleic, and finally linoleic acid should be presented; 2) Less substituted are first, 3) Less oxidized compounds are first. I could not find the order in compounds, it is not for substitution, it is not by the order of fractions purification. Am I wrong? Please choose a criteria and order the compounds if not by number in the text at least in the scheme in Fig 1.

Line 171. Where are the two bands? “Absorption maxima of 1 at 196 and 193 nm” It is not observable in Fig S3.

Line 172. Did the authors mean bonds instead of bands?

Carbonyl IR signals are strong. Please assign the band for the carbonyl group in S4. Please include in the discussion at least for reviewing purposes why the intensity of characteristic IR bands are not the same for linoleic acid and compound 1, when it has 2 linoleyl substituents.

Line 175. Capital letter after punctuation.

Please provide appropriate expansions of 1H NMR spectra and show the chemical shifts in all 1D spectra as x axis. Could the authors present the chemical shifts of each signal as group of peaks? It is difficult to read when all are overpopulated in the top of spectra.

Structures could be completely verified when authors provide appropriate expansions at least of 1D NMR spectra.

Line 192. Why the authors did not include the name of the fatty acid moiety instead of only name as acyl? “1, 2-di-O-acyl-3-O-[α-D-galactopyranosyl-(1''→6')-O-β-D-galacto-pyranosyl]-glycerol”

Line 208. Please numbered a structure for a visual reference between the structure and NMR data of table 1.

Line 215 and 253. Please check punctuation and capital letters.

Configuration assignment of compounds 1 and 2. Please provide appropriate evidence as well as discussion for the configuration assigns of compounds reported.

Section 6.2.1. Please be clear when you write “were higher”. Do you mean more potent? And more important are the values obtained from compounds statistically different from Vrp? If yes, please provide p value and mention the test used including post hoc if it is the case.

Line 292. “A representative result is shown from three independent experiments.” Please be clear. “A representative result is presented or the mean of 3 independent experiments”? Please indicate the dispersion measure you included with the error bars in all the results in the manuscript (graphs and tables.

Experimental and results section. It is not clear if ADM was assayed in combination with treatments or as independent compound, please be clear (text and figures), if the authors performed both kind of methodologies, please properly describe them.  2.5 section “Verapamil and Adriamycin were used as the positive controls and DMSO as the 143 negative control.” And 3.2.2. section The MDR reversal assay was performed using the anticancer drug adriamycin (ADM) 256 at different concentrations combined with the test compounds”.

After all the comments include here, I cannot recommend the manuscript jof-2160217 in its present form for publication. Major corrections are needed regarding methodology (experimental and description) and the species which the compounds will be reported from.

Author Response

Comments and Suggestions for Authors

Regarding the manuscript entitled ¨A Saprophytic Fungus Tubeufia rubra Produces Novel Rubracin D and E Reversing Multidrug Resistance in Cancer Cells”. Authors had conducted a chemical study including purification and chemical elucidation of 18 compounds from “whole fermented cultures” of Tubeufia rubra using the fungal strain (PF02-2) growth in PDA and PDB medium and 200 g of oat.  The authors did not specify the density of strain or how they control the content for fermentation. How the authors know that the compounds found are from the fungus and not from the oat used?

Answer: thank you very much for your comments. We performed the HRESIMS analysis of the chemical profiling of the ethyl acetate extract from the culture of strain PF02-2 and the blank culture (the oat medium without the strain).  According to the mass and MS/MS data, compounds 1-18 were just found in the ethyl acetate extract of PF02-2 stain. However, in the blank medium, the retention time, MASS and MS/MS data of compounds 1-18 were not detected. Therefore, compounds 1-18 were the metabolic products of PF02-2 stain after fermentation, not for the oat. Detailed chemical profiling of and MASS data of the extract from the culture of strain PF02-2 and the blank culture description can be found in the supplementary material.

Authors had reported 18 glyceroglycolipids and had name them rubracins as the species name T. rubra, however, other glyceroglycolipids had been reported from Avena sativa (oat) [Chemistry and Physics of Lipids 78 (1995) 97; Lipids 36, (2001) 153; Lipids 33, (1998) 355] and authors used 200 g of oat in 450 ml of total volume for fermentation. With this unique extract there is no form to be certain that the compounds are from T. rubra. Authors need more experimental work to define the origin of the compounds. There is only one previous work publishing “compounds from T. rubra” [Rec. Nat. Prod. 16:6 (2022) 622], the paper belongs to the same group and they had the same mistake (The authors cannot affirm with certainty that the metabolites isolated come from the fungus). This paper is not included in the references.

Answer: Thank you very much for your kindly comments. The origin of compound 1-18 was answered in the previous question. We have rechecked the references in the manuscript and added the reference (Rec.Nat.Prod. 16:6 (2022) 622) about the previous results from PF02-2.

Structures of compounds 1 and 2 can not be completely verified without the appropriate expansions of 1H NMR to verify multiplicity.

Answer: thank you for your thoughtful comments. We have provided the detailed information of 1H NMR proton signals including the multiplets and the coupling constants. In the supplementary materials, the expansive 1H NMR spectrum of compounds 1 was shown in Fig. S6 and 2 were also shown in Fig. S16.

Next comments should be attended too.

Answer: thank you for your careful comments. We have added the comments about multiplets and the coupling constants in the maunuscript.

Line 13. Please write with bold letters all numbers that correspond to compounds.

Answer: thank you very much for your kindly comments, we have revised in the manuscript.

Line 41. Please write all names of any species with italics.

Answer: thank you very much for your careful comments, we have revised in the manuscript.

Section 2.3. Please indicate for each chromatographic process described which kind of column were used, for example, for column chromatography and medium pressure chromatography indicate the column diameter and silica gel size or mesh, if HPLC indicate the column dimensions (long, id and particle size) and pump conditions (flow, temperature, injection vol, describe gradient if used, etc).

Answer: thank you very much for your thoughtful comments. When we rechecked purification and isolation methods from the manuscript, we found that Glyceroglycolipids (3-18) isolated from the strain Tubeufia rubra PF02-2 were not used by HPLC methods. This is our mistake, we have corrected and removed about HPLC method in our manuscript. The information about the column diameter and silica gel mesh has added to them in the manuscript.

Line 82. For convention, when work with reversed phase water is written first and then the organic solvent, you should follow the convention.

Answer: thank you for your careful comments. In the field of phytochemistry investigation, when work with reversed phase, organic solvent is written first and then water.  We also checked recent articles published in JOF (J. Fungi 8 (2022) 40; J. Fungi 8 (2022) 1058) was also written with organic solvents first, followed by water.

Lines 89, 116 and the complete manuscript. As convention, all numbers indicating number of atoms are written with subscript, please follow the convention as least for substances (line 116), for C18 it could be accepted with or without subscript, but be homogeneous, you are using two stiles (see C18 in lines 81, 89, 92). Check in the complete manuscript.

Answer: thank you for your thoughtful comments, we have revised them in the manuscript.

Lines 89-91. Please check the correct amount of each fraction in all experimental section. Authors indicated 2.6 mg for fraction 3-6-3, but 1.8 g for its subfraction 3-6-3-4.

Answer: thank you for your careful comments. We rechecked section 2.3 and found that the correct amount of Fr. 3-6-3 was 2.6 g. this is our mistake and we have corrected it in the manuscript.

Lines 128 and 132. m/z must be italics.

Answer: thank you very much for your careful comments, we have revised them of the manuscript.

Lines 126, 128, 130, 132, 141 (cells/mL). There are spaces not needed. Please check the complete manuscript.

Answer: thank you for your kindly comments. We have rechecked them and removed unnecessary spaces.

Section 2.5. Please indicate the volume and solvent/medium used for compounds tested and positive controls, as well as the concentrations used.

Answer: thank you for your thoughtful comments. We have added them to section 2.6 and 2.7 of the manuscript.

Line 157. Please check writing.

Answer: thank you very much for your kindly comments, we have rechecked and revised in the manuscript.

Lines 158, Table 2 and Fig 5. The IC50 values are a measure of the potency of compounds in specific assays models. The most common way to present IC50 is the calculation using Hill equation. I can only find information for Bliss method used for synergy, however, there are no discussion regarding synergy in the manuscript. Could the authors include a reference for the Bliss method used? Could the authors provide an explanation or discussion only for reviewing purposes the reason of showing IC50 values for each concentration assayed? Is that possible to report?  In my opinion it would be more useful if authors present the IC50 values as a single value calculated from different concentrations assayed. For that calculation at least 5 concentrations are needed, and a sigmoid curve obtained. If the authors can't do the math because of the curve shape or concentrations missing, I suggest presenting only as inhibition % with the error measure you choose (std dev, etc).

Answer: thank you for your kindly comments. Thank you very much for your thoughtful comments. We re-examined the manuscript, raw data of cytotoxic and reversed MDR activity, and we were confident that our data was complete. Each experiment was set up with 6 gradients and repeated three times. However, in the calculation of IC50 value, we used the Modified Kou's method to calculate, and its calculation formula was lgIC50= Xm-I(P-(3-Pm-Pn)/4), where Xm was the maximum dose of lg, I was lg(maximum dose/immediate dose), P was the sum of positive reaction rate, Pm was the maximum positive reaction rate, and Pn was the minimum positive reaction rate. Its IC50 value is expressed as mean ±SD. At the same time, we used SPSS 12.0 software to recalculate the IC50 value, and the results were the same as the Modified Kou's method, which proved that compounds 1 and 2 showed reversed MDR activity against the three drug-resistant tumor cell lines MCF/ADM, A549/ADM, and K562/ADM. As for the calculation method of IC50, we have corrected it in Modified Kou's method in the manuscript, and relevant raw data have been added in the supplementary materials (Fig.S21-S). This is our major mistake. We apologize for the trouble caused to your work.

Line 161. “Results are expressed as the mean standard deviation” or was expressed as the mean ± standard deviation?

Answer: thank you very much for your careful comments. The correct expression is the mean ± standard deviation, we have revised in the manuscript.

Fig 1 and the complete manuscript. Authors are using common names for fatty acid derivatives; it is suggested to use common name for octadecanoic acid too which is stearic acid.

Answer: thank you very much for your thoughtful suggestion. Common names for fatty acid derivatives are more appropriate, we have rechecked and revised according to common names for fatty acid in the manuscript.

Did the authors follow a specific order to number the compounds?

Answer: thank you very much for your careful comments. We numbered them according to new compounds combined with the sugar moieties of compounds.

Authors can improve Fig 1 following the next suggestions according to general conventions: 1) Order the substituents, first the smaller chains, first the saturated chains, thus myristic, palmitic, stearic, oleic, linoleic, and finally linoleic acid should be presented; 2) Less substituted are first, 3) Less oxidized compounds are first. I could not find the order in compounds, it is not for substitution, it is not by the order of fractions purification. Am I wrong? Please choose a criteria and order the compounds if not by number in the text at least in the scheme in Fig 1.

Answer: thank you very much for your thoughtful comments. Our numbering only considered the structure of new compounds and sugars moieties, and did not consider the fatty chain, leading to the lack of standard in Figure 1 of the manuscript. We have rearranged the order of compounds 1-18 according to your suggestion. Detailed description was shown in Fig. 1 of the manuscript. Thank you again.

Line 171. Where are the two bands? “Absorption maxima of 1 at 196 and 193 nm” It is not observable in Fig S3.

Answer: thank you for your thoughtful comments. We have rechecked UV spectra (Fig S3) and indicated that maximum absorption wavelength of 1 was only at 196 nm. We have corrected it in the manuscript.

Line 172. Did the authors mean bonds instead of bands?

Answer: thank you for your kindly comments. This is our mistake and we have changed bands to bonds in the manuscript.

Carbonyl IR signals are strong. Please assign the band for the carbonyl group in S4. Please include in the discussion at least for reviewing purposes why the intensity of characteristic IR bands are not the same for linoleic acid and compound 1, when it has 2 linoleyl substituents.

Answer: thank you for your careful comments. IR is a commonly used auxiliary tool in the structural identification of natural products. It can only be used as a qualitative research to determine the presence of characteristic functional groups of compounds. The number of functional groups was determined by combining 1D NMR, and the functional group location was determined by HMBC spectrum. In the manuscript, we can only imply the existence of hydroxyl and carbonyl functional groups from the IR of compounds 1 and 2. We have assigned the characteristic functional groups of IR spectra of compound 1 and compound 2 respectively in the supplementary material. I hope you are satisfied with my answer and thank you again.

Line 175. Capital letter after punctuation.

Answer: thank you for your thoughtful comments, we have revised in the manuscript.

Please provide appropriate expansions of 1H NMR spectra and show the chemical shifts in all 1D spectra as x axis. Could the authors present the chemical shifts of each signal as group of peaks? It is difficult to read when all are overpopulated in the top of spectra.

Answer: thank you for your thoughtful comments. We have added the extended 1H-NMR spectrum of compounds 1 and 2 in the supplementary material, through which the structure of compounds 1 and 2 can be better displayed in the supplementary material.

Structures could be completely verified when authors provide appropriate expansions at least of 1D NMR spectra.

Answer: thank you very much for your kindly comments. We have added the extended 1D NMR spectrum of compounds 1 and 2 in the supplementary material.

Line 192. Why the authors did not include the name of the fatty acid moiety instead of only name as acyl? “1, 2-di-O-acyl-3-O-[α-D-galactopyranosyl-(1''→6')-O-β-D-galacto-pyranosyl]-glycerol”

Answer: thank you very much for your careful comments. In the process of structure identification of compound 1, the first part focuses on the carbon and hydrogen signals of sugar and glycerol. Compound 1 contains two sugar moieties and the structure of glycerol are determined by the chemical shift of hydrogen and the coupling constant combined with 13C-NMR spectra. At this time, the name of the compound could not include fatty acid moieties and could only be 1,2-di-O-acyl-3-O-[α-D-galactopyranosyl-(1"→6")-O-β-D-Galacto-Pyranosyl]-glycerol. The latter part focuses on the carbon and hydrogen signals of fatty acid moieties and acetyl group.

Line 208. Please numbered a structure for a visual reference between the structure and NMR data of table 1.

Answer: thank you for thoughtful comments. We have added the numbered compounds 1 and 2 to the manuscript.

Line 215 and 253. Please check punctuation and capital letters.

Answer: thank you for kindly comments, we have rechecked and revised in the manuscript.

Configuration assignment of compounds 1 and 2. Please provide appropriate evidence as well as discussion for the configuration assigns of compounds reported.

Answer:  thank you for your thoughtful comments. We added the configuration assignment of compounds 1 and 2 the manuscript.

Section 6.2.1. Please be clear when you write “were higher”. Do you mean more potent? And more important are the values obtained from compounds statistically different from Vrp? If yes, please provide p value and mention the test used including post hoc if it is the case.

Answer: thank you for your careful comments. As the test compound and positive control Vrp were the same dose for three MDR cells during the experimental design and detailed description was shown in the section 2.7 of the manuscript. Therefore, in the manuscript, we can infer the reversed MDR activity of the tested compounds (1-2) and Vrp by comparing the IC50 value.

Line 292. “A representative result is shown from three independent experiments.” Please be clear. “A representative result is presented or the mean of 3 independent experiments”? Please indicate the dispersion measure you included with the error bars in all the results in the manuscript (graphs and tables.

Answer: thank you for your kindly comments. In the experimental design, we set 6 concentration gradients for each experiment and each experiment was performed in triplicate, and IC50 was expressed in the form of mean ±SD. I apologize for our unclear description. and we have revised in the manuscript. I hope you can understand.

Experimental and results section. It is not clear if ADM was assayed in combination with treatments or as independent compound, please be clear (text and figures), if the authors performed both kind of methodologies, please properly describe them.  2.5 section “Verapamil and Adriamycin were used as the positive controls and DMSO as the 143 negative control.” And 3.2.2. section The MDR reversal assay was performed using the anticancer drug adriamycin (ADM) 256 at different concentrations combined with the test compounds”.

Answer: thank you very much for your thoughtful comments, we have added detailed cytotoxicity assays in section 2.5 and section 2.6 about reversed MDR assay in the manuscript.

After all the comments include here, I cannot recommend the manuscript jof-2160217 in its present form for publication. Major corrections are needed regarding methodology (experimental and description) and the species which the compounds will be reported from.

Reviewer 3 Report

The article describes the study of Two new (1, 2 viz Rubracin D and E) and sixteen known (3-18) Glyceroglycolipids in the 13 saprophytic fungus Tubeufia rubra (PF02-2). The investigation reported in the paper is interesting. However, before subjecting it to the publication, its modification in light of following comments is needed.

1. Part of Introduction very shortly, Write more about secondary metabolites isolated from other fungi and their biological activities

2. Experimental details given are inadequate. How much fungal culture was extracted to afford 2027.17 g of crude extract? What was the volume of the EtOAc to extract this quantity of the culture? How the extract was isolated?

3. Detailed analytical conditions for analytical and preparative HPLC should be included. A superficial account of these experiments has been even which is not acceptable.

4. In the Supplementary material section Physical conditions of each of the known 16 isolated compounds should be given. Characterization of each of these compounds should be discussed. Since molecular mass and NMR data of these compounds was not determined, how their identity can non ambiguously be established?

5. Add these two studies to support this sentence with recent reference

“Natural products play a key role in developing new clinic drug because of their (Line 36) diverse structures and potential biological activities (Line 37)”

https://www.mdpi.com/1422-0067/23/19/11935

https://www.tandfonline.com/doi/abs/10.1080/22311866.2022.2154265

6. In the Supplementary material section, write the figure captions underneath each figure

7.  Authors should measure the cytotoxic activity of 16 known compounds also.

8. Authors should be measure the selectivity index of 2 new compounds against K562/ADM and A549/ADM cancer cells

8. Language also needs improvement.

Author Response

The article describes the study of Two new (1, 2 viz Rubracin D and E) and sixteen known (3-18) Glyceroglycolipids in the 13 saprophytic fungus Tubeufia rubra (PF02-2). The investigation reported in the paper is interesting. However, before subjecting it to the publication, its modification in light of following comments is needed.

  1. Part of Introduction very shortly, Write more about secondary metabolites isolated from other fungi and their biological activities.

Answer: thank you very much for your careful comments, we have revised the introduction section of the manuscript.

  1. Experimental details given are inadequate. How much fungal culture was extracted to afford 2027.17 g of crude extract? What was the volume of the EtOAc to extract this quantity of the culture? How the extract was isolated?

Answer: thank you very much for your careful comments. We have added the volume of the EtOAc and this quantity of the fermented cultures in line 90 and 92 of the manuscript. The extract separation process is described in lines 94-139 of the manuscript.

  1. Detailed analytical conditions for analytical and preparative HPLC should be included. A superficial account of these experiments has been even which is not acceptable.

Answer: thank you very much for your thoughtful comments. When we rechecked purification and isolation methods from the manuscript, we found that Glyceroglycolipids (3-18) isolated from the strain Tubeufia rubra PF02-2 were not used by HPLC methods. This is our mistake, we have corrected and removed about HPLC method in our manuscript.

  1. In the Supplementary material section Physical conditions of each of the known 16 isolated compounds should be given. Characterization of each of these compounds should be discussed. Since molecular mass and NMR data of these compounds was not determined, how their identity can non ambiguously be established?

Answer: thank you for your kindly comments. 1D NMR and HR-ESI-MS data of compounds (318) was added to supplementary material (see Table S1). According to the comparison with the reported NMR data and mass spectra in the literature, compounds (318) were identified. The relevant references were also provided in the manuscript.

  1. Add these two studies to support this sentence with recent reference

“Natural products play a key role in developing new clinic drug because of their (Line 36) diverse structures and potential biological activities (Line 37)”

https://www.mdpi.com/1422-0067/23/19/11935

https://www.tandfonline.com/doi/abs/10.1080/22311866.2022.2154265

Answer: thank you very much for your suggestion. We have added them in our manuscript.

  1. In the Supplementary material section, write the figure captions underneath each figure

Answer: thank you for your careful comments. We have added figure captions in the supplementary material section..

  1. Authors should measure the cytotoxic activity of 16 known compounds also.

Answer: Thank you very much for your careful comments. After purification and structural analysis of the secondary metabolites from the strain Tubeufia rubra, we firstly focused on finding bioactive compounds with novel structures. Therefore, we screened the cytotoxic activity of the new compounds (1-2) against three strains of drug-resistant tumor cells, MCF-7/ADM, K562/ADM and A549/ADM. The results showed that compound (1-2) had no cytotoxic activity against three strains of drug-resistant tumor cells at an effective concentration below 25 μg/mL. It can be further applied to reversed MDR activity studies, which have been described in detail in the manuscript. As for known compounds (3-18), many literatures have reported their activities, such as anti-inflammatory (Arch Pharm. Res. 35 (2012) 2135; J Agric. Food Chem. 61 (2013) 7081), antitumor (Bioorganic Chemistry 78 (2018) 381), antifungal (Natural Product Communications, 8 (2013) 1285), etc., so we do not perform the activity screening of known compounds (3-18). We promise that we will consider the activity screening of known compounds in the future studies, please understand.

  1. Authors should be measure the selectivity index of 2 new compounds against K562/ADM and A549/ADM cancer cells.

Answer: thank you for your thoughtful comments. The cytotoxicity against K562/ADM and A549/ADM cancer cells of new compounds was shown in section 3.2.1 of the manuscript and we have added the raw data of cytotoxicity against K562/ADM and A549/ADM to the supplementary material (Fig.S26 and S31).

  1. Language also needs improvement.

Answer: thank you for your kindly comments. We have asked Professor Cheng to help us with the language modification.

Round 2

Reviewer 3 Report

The authors revised the manuscript satisfactorily, I recommend the manuscript to be published in this journal